# Protection against Glucolipotoxicity by High Density Lipoprotein in Human PANC-1 Hybrid 1.1B4 Pancreatic Beta Cells: The Role of microRNA

**DOI:** 10.3390/biology10030218

**Published:** 2021-03-13

**Authors:** Jamie M.R. Tarlton, Richard J. Lightbody, Steven Patterson, Annette Graham

**Affiliations:** Department of Biological and Biomedical Sciences, School of Health and Life Sciences, Glasgow Caledonian University, Glasgow G4 0BA, Scotland, UK; Jamie.Tarlton@gcu.ac.uk (J.M.R.T.); Richard.Lightbody@gcu.ac.uk (R.J.L.); Steven.Patterson@gcu.ac.uk (S.P.)

**Keywords:** glucolipotoxicity, beta-cells, high density lipoprotein, microRNA, hsa-miR-21-5p

## Abstract

**Simple Summary:**

Loss of the cells which secrete insulin, the hormone which controls blood sugar levels, can occur due to the damaging effects of high levels of sugar and fat in the bloodstream, and is a key factor which triggers type 2 diabetes. Protection against this damage is provided by one of the fat and protein complexes which carry the ‘good’ cholesterol in the bloodstream, a complex which is called high-density lipoprotein. This study looks at the changes in the expression of small pieces of RNA, called microRNA, triggered in insulin-secreting cells by treatment with high density lipoprotein. These small RNA sequences can help to regulate the expression of genes which can contribute to the protective effect of high-density lipoproteins. Delivery of one of these sequences was able to modulate gene expression in cells, but could not provide enough protection against the damaging effects of high sugar and fat. We conclude that the complex changes in numerous microRNA sequences caused by high density lipoproteins cannot be replicated by a single sequence, suggesting that networks of regulatory microRNA sequences are more important than single sequences. The results will inform drug discovery strategies which focus on high density lipoprotein and microRNA.

**Abstract:**

High-density lipoproteins provide protection against the damaging effects of glucolipotoxicity in beta cells, a factor which sustains insulin secretion and staves off onset of type 2 diabetes mellitus. This study examines epigenetic changes in small non-coding microRNA sequences induced by high density lipoproteins in a human hybrid beta cell model, and tests the impact of delivery of a single sequence in protecting against glucolipotoxicity. Human PANC-1.1B4 cells were used to establish Bmax and Kd for [^3^H]cholesterol efflux to high density lipoprotein, and minimum concentrations required to protect cell viability and reduce apoptosis to 30mM glucose and 0.25 mM palmitic acid. Microchip array identified the microRNA signature associated with high density lipoprotein treatment, and one sequence, hsa-miR-21-5p, modulated via delivery of a mimic and inhibitor. The results confirm that low concentrations of high-density lipoprotein can protect against glucolipotoxicity, and report the global microRNA profile associated with this lipoprotein; delivery of miR-21-5p mimic altered gene targets, similar to high density lipoprotein, but could not provide sufficient protection against glucolipotoxicity. We conclude that the complex profile of microRNA changes due to HDL treatment may be difficult to replicate using a single microRNA, findings which may inform current drug strategies focused on this approach.

## 1. Introduction

The ‘pre-diabetic’ condition is characterised by insulin resistance, which eventually triggers beta cell failure, loss of compensatory insulin secretion and the onset of frank type 2 diabetes. Beta cell damage can be induced by glucolipotoxicity (GLT) [1], which describes the deleterious effects of elevated levels of free fatty acids (FFA) on glucose homeostasis, resulting in concurrent, and synergistically damaging, exposure of beta-cells to high glucose levels. Pleiotropic outcomes triggered in beta cells by high glucose/FFA conditions, include oxidative stress, mitochondrial dysfunction, endoplasmic reticulum (ER) stress, inflammation, impaired autophagy and loss of glucose-stimulated insulin secretion, reflecting alterations in multiple cell signalling pathways [2], enhanced expression of pro-inflammatory, lipogenic and pro-apoptotic genes and proteins [3], and accumulation of lipids such as ceramides, cholesterol and cholesteryl esters and triacylglycerols [2,3,4].

By contrast, high-density lipoprotein (HDL) [5,6,7,8], its major apolipoprotein (apo), apoA-I [7,9,10,11], and/or the ATP binding cassette (ABC) transporters with which they interact [12,13,14,15,16], can provide protection to beta cells and pancreatic islets in experimental models, findings generally supported by clinical data [17,18,19,20,21]. Studies variously link these protective functions with sterol removal or efflux from beta cells [8,12,13,14,15,16]; others report cholesterol- and/or transporter-independent effects of apoA-I and HDL [7,10], while conflicting reports exist regarding beta cell function in carriers of loss of function (LOF) mutations in ABCA1 [22,23]. Some of these divergent outcomes may be attributed to the heterogeneous nature of HDL and its cargo molecules [24,25], and to the widely differing concentrations of apoA-I (10 μg protein mL^−1^ to 900 μg mL^−1^) and/or HDL (50 μg protein mL^−1^ to 1 mg mL^−1^) used, or required, to demonstrate the protective effect of this lipoprotein in human and rodent beta cell models (reviewed in [5]).

Certainly, numerous cell signalling pathways are altered in beta cells by exposure to apoA-I and HDL, including G protein coupled receptor (GPCR) activation of adenyl cyclase, protein kinase A and forkhead box protein 01 (Fox01) [9], calcium signalling [7], and activation of hedgehog (Hh) receptor Smoothened (SMO) [8], protein kinase B [26] and sphingosine-1-phosphate receptors (S1PR1-3) [5,24,27]. Transcription factors, such as the Liver X receptor (LXR), peroxisome proliferator activator receptor alpha (PPARα), FOXO1 and BTB domain and CNC homolog 1 (BACH1) have also emerged as key factors orchestrating cellular responses to fatty acids such as palmitate [3]. Further, in the last decade, there has been an explosion of interest in the epigenetic mechanisms, including the role of microRNA sequences, involved in beta cell (dys)function and type 2 diabetes [28]. MicroRNA are small, non-coding (~22 nucleotide) sequences which can regulate networks of protein expression by binding to complementary mRNA binding regions, resulting in either mRNA degradation or translational inhibition [28].

However, the epigenetic mechanisms that contribute to the protective effects of apoA-I and HDL in human beta cells remain relatively uncharacterised. This study establishes, by microchip array, the microRNA signature associated with apoA-I and HDL treatment in human hybrid (1.1B4) pancreatic beta cells, at concentrations of these acceptors that induce cholesterol efflux; after validation of the microchip array, the ability of a single microRNA sequence, hsa-miR-21-5p, to protect against GLT was then investigated in the presence or absence of HDL.

## 2. Materials and Methods

### 2.1. Cell Culture and Maintenance

Human PANC-1.1B4/human beta cell hybrid 1,1B4 cells (1.1B4 cells) were obtained from Public Health England Culture Collections (#10012801). This cell line was generated by electrofusion of human pancreatic beta cells and the immortal human PANC-1 epithelial cell line, characterised by McCluskey et al. (2011) [29], and recently employed by Nemecz et al. (2019) [30] to study glucolipotoxicity, and by Honkimaa et al. (2020) to study genetic adaptation of Coxsackievirus B1 [31]. This human cell line was selected for this study as human and rodent microRNA sequences and targets are not always conserved; however, please note that this cell line has recently been discontinued (ECACC) due to possible contamination with rat DNA (https://www.phe-culturecollections.org.uk/products/celllines/discontinued-ecacc-cell-lines/11b4.aspx accessed on 20 January 2021).

The cells were maintained in ‘complete’ RPMI 1640 media (Lonza, Basel, Switzerland) containing foetal bovine serum (FBS, 10%, *v*/*v*; Gibco, Thermo Fisher Scientific, Waltham, MA, USA)and penicillin/streptomycin (50 U mL^−1^; 50 μg mL^−1^, respectively; Gibco, Thermo Fisher Scientific, Waltham, MA, USA), at 37 °C and 5% CO_2_. Cells were sub-cultured at 80% confluency at a ratio of 1:4 to 1:8, every 3–4 days, and utilised between passages 29 and 42. Experiments using apo(lipo)proteins, apoA-I and high-density lipoprotein (HDL) (Athens Research & Technology, Athens, GA, USA), were performed in serum-free RPMI media supplemented with penicillin/streptomycin, as described above.

### 2.2. Cellular Lipid Extraction and Analysis

Cholesterol efflux (0–24 h; 2.5 × 10^5^ cells/well in a 12-well plate) was determined in serum-free media, in the presence and absence of apoA-I (0.5–20 μg mL^−1^) or HDL (0.5–80 μg mL^−1^) in cells previously radiolabelled with [^3^H]cholesterol (1 μCi mL^−1^; Perkin Elmer, Beaconsfield, UK) for 20 h, and then equilibrated (18 h) in media containing 1.0% (*w*/*v*) fatty acid free bovine serum albumin (BSA; Sigma Aldrich, Gillingham, UK). Lipids were extracted from cells using hexane:isopropanol (3:2, *v*/*v*) and dried at 37 °C under a stream of N_2_. Efflux was calculated as % Efflux = (dpm media)/(dpm media + dpm cells) × 100%, as reported previously [32].

### 2.3. Glucolipotoxicity Assays

A working solution of 5mM palmitic acid conjugated with 10% (*w*/*v*) fatty acid free BSA was diluted to the appropriate concentration in serum-free RPMI 1640, and glucose added from a 1 M stock to achieve a final concentration of 30mM. Cell viability (2.5 × 10^4^ cells/well in a 96-well plate) was measured by conversion of thiazolyl blue tetrazolium bromide to formazan [33] and using the PrestoBlue cell viability assay (Thermo Fisher Scientific, Waltham, MA, USA) [34]. CellEvent Caspase-3/7 Green Detect Reagent (Thermo Fisher Scientific, Waltham, MA, USA) was used to monitor apoptosis [35].

### 2.4. Total RNA Isolation, Microchip Analysis and Q-PCR

Total RNA (5 × 10^5^ cells/well in a 12-well plate) was collected using either a Direct-zol RNA MiniPrep Plus kit (Zymo Research, Cambridge, UK) or a Nucleospin RNA kit (Machery-Nagel, Düren, Germany). Complementary DNA (cDNA) used for measurement of gene expression (mRNA) was generated from 250 ng RNA using the High-Capacity cDNA Reverse Transcription kit (Applied Biosystems, Thermo Fisher Scientific, Waltham, MA, USA) according to manufacturer’s instructions; cDNA used to measure miRNA and/or mRNA was generated using the miScript II RT kit (Qiagen, Manchester, UK) and HiFlex buffer, as indicated by the manufacturer.

Microarray analysis of miRNA expression was performed commercially by LC Sciences (Houston, TX, USA); 2632 unique mature human miRNA sequences derived from version 22 of miRBase [36] were measured using μParaflo microfluidic chip technology (https://www.lcsciences.com/discovery/applications/transcriptomics/mirna-profiling/mirna/ accessed on 3 January 2018), exemplified by Zhang et al. (2019) [37]. Relative quantitative PCR (Q-PCR) for miRNA and mRNA expression was performed using the primer sequences defined in Appendix A using HOT FIREPol EvaGreen qPCR Mix Plus (Solis BioDyne) on a C1000 thermal cycler with a CFX96 real-time system attached; melt curves were performed for all primers and none evidenced dimer formation. The expression of target miRNA and mRNA was determined using the 2^−ΔΔ^Ct method relative to an invariant control sequence/gene, as indicated in legends to figures; statistical analyses were performed using the ΔCt values, compared with the relevant housekeeping sequence.

### 2.5. Transfection with miRNA Mimics and Inhibitors

Transfection of 1.1B4 cells (2 × 10^5^ cells/well in a 12-well plate) with the hsa-miR-21-5p mimic, and scrambled mimic control (Qiagen), and the miR-21-5p inhibitor and scrambled inhibitor control (Qiagen) was achieved using HiPerFect Transfection Reagent (Qiagen), according to the manufacturer’s instructions. In brief, transfection complexes (100 μL) were added in serum-free RPMI 1640 and incubated for 4h prior to the addition of further serum-free RPMI 1640 (400 μL). The cells were incubated at 37 °C and 5% CO_2_ with the concentrations of mimic and inhibitor, and for the periods of time, indicated in figure legends.

### 2.6. Bioinformatic Analysis

MicroRNA sequences differentially expressed after treatment with apoA-I and HDL, compared with the basal control, were analysed using DNA Intelligent Analysis (DIANA)-miRPath v3.0 [38,39] to identify pathways with *p* < 0.01 association with differentially expressed genes. Upregulated and downregulated miRNAs were analysed separately in miRPath; this utilized experimentally validated miRNA interactions in DIANA-TarBase v7.0 database [35] incorporated into Kyoto Encyclopedia of Genes and Genomes (KEGG) [40] pathways to identify the most closely associated functions with each group. Disease-specific pathways were excluded from the results. Interaction networks were created using Cytoscape v3.8.0 [41]. The graphical abstract was created using Biorender.com.

### 2.7. Statistical Analysis

All datasets and methodologies were deposited in Mendeley Data (doi:10.17632/s3bznmgtpg.1). Results are expressed as means ± SD or SEM of the number of experiments indicated in legends to figures. Statistical analysis was performed by one-way, two-way, or three-way ANOVA and Dunnett’s post-test, repeated measures ANOVA and Dunnett’s post-test, or Student’s t test, as indicated in legends to figures. All statistical testing was performed using Graph Pad Prism software; * *p* < 0.05; ** *p* < 0.01 and *** *p* < 0.001.

## 3. Results

### 3.1. Cholesterol Efflux from 1.1B4 Cells

Functional responses to apoA-I and HDL in 1.1B4 cells were established by measurement of [^3^H]cholesterol efflux to these acceptors, shown in Figure 1. Efflux to apoA-I (Figure 1A) was relatively low (estimated Bmax = 0.99 ± 0.15%) and plateaued (Kd = 3.70 ±1.84 μg mL^−1^), with significant increases in efflux above the control condition noted at 10 and 20 μg mL^−1^. By contrast, efflux to HDL (Figure 1B) exhibited a curvilinear relationship, with a substantially higher estimated Bmax (37.75 ± 3.69%) and Kd (77.66 ±12.78 μg mL^−1^); concentrations of HDL (5–80 μg mL^−1^) significantly increased [^3^H]cholesterol efflux above control. The levels of mRNA for ABCA1, which effluxes cholesterol to apoA-I, and of ABCG1, which effluxes to HDL, were measured: gene expression (2^−ΔCt^; mean ±SEM) for ABCA1, relative to PPIA, was 2.01 × 10^−5^ ± 1.25 × 10^−5^ (*n* = 3), while levels of ABCG1 relative to PPIA were 4.77 × 10^−5^ ± 3.6 × 10^−5^ (*n* = 3); equivalent primer efficiencies were noted here (102% and 97%, respectively). These values are not significantly different, suggesting that the activity and stability of these proteins contribute to the efflux outcomes observed. Over time, [^3^H]cholesterol efflux to apoA-I (20 μg mL^−1^) and HDL (20 μg mL^−1^) adhered strictly to linearity (Figure 1C) with highly significant (*p* < 0.001) differences noted between the slope of each curve.

### 3.2. Glucolipotoxicity in 1.1B4 Cells

The impact of treatment with high glucose (30 mM) combined with a palmitic acid (PA)/BSA complex (0.1 mM to 0.5mM PA) on the viability of 1.1B4 cells over 24 h is shown in Figure 2A. No significant changes in viability due to any of these treatments were noted at 0 h, as judged by conversion of MTT to formazan (data not shown); however, significant reductions in viability were noted after 24 h at concentrations of 0.25 mM PA (43.4%; *p* < 0.05) and 0.5 mM PA (72.6%; *p* < 0.01), compared with the control incubation. The effect of high glucose in combination with 0.25 mM PA on 1.1B4 cell viability was also confirmed via cellular reduction of resazurin to fluorescent resorufin (47%; *p* < 0.001) (Figure 2B). Addition of apoA-I at concentrations of 40 and 80 μg mL^−1^ (8 μM to 16 μM) significantly increased cell viability, as judged by conversion of MTT to formazan, by 1.15-fold (*p* < 0.05) and 1.19-fold (*p* < 0.01), respectively. However, delivery of ApoA-I at 20–80 μg mL^−1^ did not protect against glucotoxicity enhanced by 0.25 mM PA (Figure 2C). By contrast, the addition of HDL at 40 and 80 μg mL^−1^ (~0.05 mM to 0.1 mM HDL-cholesterol) increased conversion of MTT to formazan by 1.68-fold (*p* < 0.05) and 2.33 fold (*p* < 0.001), compared with the GLT challenge alone (Figure 2D).

Further, in the more sensitive resorufin cell viability assay, 20 μg protein mL^−1^ HDL (~0.025 mM HDL-cholesterol) significantly improved cell viability by 1.38-fold (*p* < 0.01) under the same conditions (Figure 3A). Accordingly, this lowest concentration of HDL was tested for its impact on cellular apoptosis (Figure 3B): analysis by two-way ANOVA indicate a significant effect of time in all four conditions tested (*p* < 0.001) and no significant overall effect of HDL. There was a significant interaction between time and HDL treatment (*p* < 0.001), with HDL significantly reducing apoptosis due to GLT challenge, under serum-free conditions.

### 3.3. Changes in microRNA Expression Due to apoA-I and HDL in 1.1B4 Cells

Figure 4 shows the changes in microRNA expression detected by a global, unbiased microchip analysis after exposure of 1.1B4 cells to 20 μg mL^−1^ ApoA-I, or the equivalent concentration of HDL, for 24 h; the full report of all sequences analysed for Figure 4 has been deposited in Mendeley (doi:10.17632/s3bznmgtpg.1). Sequences common to both treatments include significant elevations in hsa-miR-25-3p and hsa-miR-29a-3p, and reductions in hsa-miR-30e-5p, hsa-miR-146a-5p and hsa-miR-335-3p. Equally, a number of miR sequences appear regulated selectively by HDL treatment, including increased expression of hsa-miR-21-5p and hsa-miR-7977, and loss of expression of hsa-miR-15b-5p, hsa-miR-16-5p, hsa-miR-130a-3p and hsa-miR-191-5p.

The elevation of hsa-miR-21-5p in 1.1B4 cells detected by microchip analyses was of particular interest as this sequence and it targets have been associated with increased proliferation and reduced apoptosis [42,43,44,45,46], beta cell number and function [47,48,49] and the aetiology of diabetes [50,51,52,53,54]. Consequently, HDL-induced changes in miR-21-5p expression noted in the microchip array were confirmed by Q-PCR (Figure 5A): using this methodology, an increase due to HDL (63%; *p* < 0.05), but not apoA-I, was noted. Treatment with HDL also reduced expression of two out of five predicted (miRBase 22) gene targets of miR-21-5p, *ZNF367* (87%; *p* < 0.05), *SMAD7* (33%; *p* < 0.05) and *STAT3* (23%; *p* < 0.05) (Figure 5B), but not *PDCD4* or *FOXO3*. Transient delivery (24h) of a miR-21-5p mimic (5nM) to the 1.1B4 cell line (Figure 5C) achieved a 37% (*p* < 0.01) increase in levels of this sequence, and significant reductions in expression of *SMAD7* (26%; *p* < 0.05) and *STAT3* (31%; *p* < 0.01), but not of the other target genes investigated. By contrast, delivery of a miR-21-5p inhibitor (10 nM) achieved a 21% (*p* < 0.05) reduction in the expression of miR-21-5p, but did not significantly increase target gene expression (Figure 5D).

### 3.4. Hsa-miR-21-5p and Glucolipotoxicity in 1.1B4 Cells

The effect of the miR-21-5p mimic and inhibitor on the viability of 1.1B4 cells, in the presence and absence of HDL, is shown in Figure 6. Transfection with the control, mimic or inhibitor did not alter cell viability compared with the wild type cells; neither the mimic (Figure 6A) or the inhibitor (Figure 6B) could alter the impact of the GLT challenge on the viability of 1.14 cells, compared with the relevant control. The effect of the miR-21-5p inhibitor, in the presence of HDL in the GLT challenge, is shown in Figure 6C; the protective effect of HDL was sustained under transfection conditions, but was not altered by the presence of the miR-21-5p inhibitor, despite the established anti-apoptotic effects of this miRNA sequence [28].

### 3.5. Pathways Targeted by microRNA Sequences Altered by HDL Treatment

Bioinformatic analysis via miRPath v3.0 showed that 18 KEGG pathways were associated with miRNAs downregulated by 20 µg mL^−1^ HDL (hsa-miR-30e-5p, hsa-miR-146a-5p, hsa-miR-130a-3p, hsa-miR-181a-5p, hsa-miR-15b-5p, hsa-miR-335-5p, hsa-miR-425-5p, hsa-miR-16-5p, hsa-miR-340-5p and hsa-miR-191-5p) (Figure 7A) and 13 KEGG pathways were associated with upregulated miRNAs (hsa-miR-7977, hsa-miR-25-3p, hsa-miR-29a-3p and hsa-miR-21-5p) (Figure 7B). miR-7977 did not contribute to the analysis as there are no interactions documented in TarBase. Pathways such as fatty acid biosynthesis and metabolism were common to downregulated miRNAs, and upregulated by treatment with HDL. Nine KEGG pathways were unique to downregulated miRNAs, including the TGF-beta signalling pathway (*p* = 3.71 × 10^−9^) and protein processing in the endoplasmic reticulum (*p* = 2.48 × 10^−7^). Four pathways were unique to miRNAs upregulated by HDL, including the FoxO signalling (*p* = 1.99 × 10^−3^) and fatty acid elongation (*p* = 7.43 × 10^−5^) pathways.

## 4. Discussion

This study demonstrates the protective effects of HDL, at concentrations below those required for transporter saturation but which sustain linear cholesterol efflux, in 1.1B4 cells subject to a GLT challenge. These well-characterised conditions induce a network of changes in miRNA sequences, as determined by microchip array, including increased expression of miR-21-5p which mediates anti-apoptotic effects [28]. Delivery of a miR-21-5p mimic replicated the impact of HDL on gene expression of *STAT3* (signal transducer and activator of transcription 3) and *SMAD7* (decapentaplegic protein 3), two established targets of this sequence; no changes in expression of *STAT3* and *SMAD7* were induced by delivery of miR-21-5p inhibitor. However, neither the mimic nor the inhibitor, the latter in the presence or absence of HDL, could significantly change the response of 1.1B4 cells to a GLT challenge. The data suggest that the complex epigenetic changes induced by HDL in human beta cells is unlikely to be replicated by a single miRNA sequence, supporting the emerging concept that global changes in miRNA networks may be more biologically relevant than changes in a single sequence [55].

Exposure to HDL has previously been linked with epigenetic changes in microRNA in a number of differing cell types [56], but this is the first report of the impact of these (apo)lipoproteins in an in vitro human beta cell model; equally, a number of microRNA sequences have been linked with (dys)function of beta cells [28]. Examination of these literature, compared with the microchip array reported here (Figure 4) reveal some common sequences [28,47,48,55,56,57,58]. For example, miR-21 has been shown to target Snc2, decreasing *Vamp2* and GSIS, but also to significantly modulate apoptosis in a range of rodent and human islet cell lines and tissues [28,48,49]. Mir-29 has been linked with increases in insulin exocytosis, but also with increased apoptosis via reductions in *Mcl1* [28]; miR-130 and miR-335 exert opposing actions on GSIS, while miR-146a is thought to increase apoptosis by increased expression of *c-Jun* [28,47,48,49]. However, treatment with HDL also alters expression of other sequences, not previously linked to maintenance of beta cell function [55,59]. Some have no function ascribed to them at present (hsa-miR-15b-5p, hsa-miR-181a-5p, hsa-miR-426-5p) while hsa-miR-16-5p was found to be down-regulated after bariatric surgery in obese patients [49], and hsa-miR-191-5p emerged as differentially regulated in Alzheimer’s disease and major depressive disorder [58].

Delivery of the miR-21-5p mimic significantly altered the level of this sequence in 1.1B4 cells, to a similar extent as HDL treatment, exerting comparable reductions in gene expression of its established target genes *STAT3* and *SMAD7*, indicating that the mimic was effective in this cell type. High density lipoprotein has previously been linked with activation of the pro-survival STAT3 signalling pathway [59,60], which protects against inflammation and apoptosis in many cell types [59,60,61,62,63]. However, Zeng et al. (2017) found that inhibition of expression and phosphorylation of STAT3 in hepatic cells restored insulin sensitivity in mice fed a high fat diet [64], while inhibition of the STAT3 pathway has also been linked with reductions in murine arterial atherosclerosis [65]. Repression of *SMAD7* has not previously been reported after HDL treatment, but this protein is a well-established inhibitor of the TGFβ signalling pathway which regulates fibrosis, inflammation and wound healing and cancer, and loss of *Smad7* in mice is associated with enhanced cell proliferation and reduced apoptosis [66]. Thus, it is clear that multiple molecular mechanisms may underlie the overall impact of HDL on apoptosis [28].

Indeed, the function of miR-21-5p in beta cells also needs careful consideration. Many studies indicate an anti-apoptotic role for this sequence in differing cell types and via a variety of mechanisms [42,43,44,45,46]. In particular, knockdown of miR-21 increases translation of proinflammatory tumour suppressor programmed cell death protein 4 (PDCD4), restricts cell proliferation and promotes apoptosis, while high levels of miR-21 are associated with cancer [42]. It is therefore possible that the elevation in miR-21-5p due to HDL may contribute to survival of 1.1B4 cells, via a mechanism which is not affected by gene repression of *STAT3* but could be aided by reduction in expression of *SMAD7* (Figure 5). However, it is clear that the transient (and modest) increase in miR-21-5p levels due to the presence of the mimic alone was not sufficient to prevent the loss of viability due to GLT challenge, and nor could the inhibitor block the protective impact of HDL in this cell line under the same conditions (Figure 6). Ruan et al. (2011) reported that elevated levels in miR-21, which induces a deficiency in PDCD4, can inhibit nuclear factor-kB (NF-κB) and protect beta-cells against streptozotocin-induced cell death, and spontaneous autoimmune diabetes in non-obese diabetic mice and C57BL7 mice [57,58]. Pro-inflammatory cytokines induce the expression of miR-21 [48], and Backe et al. (2014) suggested this is a protective response to preserve beta cell viability [47]. The feasibility of miR-21-5p as a pro-survival therapeutic agent, however, has been cast into doubt by the finding that stable (long-term) expression of miR-21-5p not only increased the proliferation of rat INS-1 insulinoma cells, but also stimulated nitric oxide synthesis and apoptosis [28,64].

Analysis by miRPath established numerous KEGG pathways associated with miRNAs differentially expressed in 1.1 beta cells following exposure to HDL (Figure 7). Three pathways that were linked to mRNAs down- or upregulated by HDL were fatty acid biosynthesis, fatty acid metabolism and lysine degradation, previously linked to the role of HDL in reverse cholesterol transport [67]. Additional pathways, including the Hippo signalling pathway, extracellular matrix (ECM)-receptor interactions and thyroid hormone signalling suggest a role for HDL in regulating developmental processes in beta cells. It is understood that miRNAs are involved in maturation of beta cells and maintenance of beta cell identity with miR-7, miR-9, miR-375, miR-376, miR-17-5p and miR-29b-3p examined in this context [68,69]. In conjunction with identification of adherens junction, cell cycle and p53 signalling pathways, this suggests that HDL modulate pathways in beta cells that control responses to stress, cell proliferation and apoptosis, supporting our observations that HDL protect 1.1B4 beta cells from damage due to GLT challenge. Two KEGG pathways uniquely linked to miRNAs upregulated by HDL were also associated with miR-21-5p (Figure 8): fatty acid elongation and the FoxO signalling pathway. Notably, the FoxO signalling pathway integrates a number of signalling pathways such as TGF-beta, PI3K-Akt and Jak-STAT, to regulate functions including cell metabolism, cell cycle regulation and apoptosis [70,71].

If we consider the limitations of this study, there are several points to note. One key point is that the study is restricted to use of the 1.1B4 cell line (with the associated drawbacks discussed above): replication in primary human islet beta cells would provide greater impact to the findings. Further, the use of metabolic activity assays (MTT and resorufin conversion) as measures of cell viability, and of caspase 3/7 activity as a measure of apoptosis, do not provide insight into the mechanism(s) by which HDL protect the 1.1B4 cells from a GLT challenge. Finally, the analysis of microRNA sequences altered by HDL treatment, compared with the control, derive from single samples using three technical replicates on each microchip, albeit validated in three independent experiments by Q-PCR.

## 5. Conclusions

This study observed that low concentrations of HDL protect 1.1B4 beta cells from the reduction in cell viability, and increased apoptosis, induced by challenge with 30 mM glucose and 0.25 mM palmitic acid (GLT). Analysis by microchip array following exposure to HDL demonstrated differentially expressed miRNAs which were associated with KEGG pathways linked with beta cell development, cell proliferation and apoptosis. Hsa-miR-21-5p upregulation by HDL was validated by qPCR and shown to lead to downregulation of *SMAD7* and *STAT3*. Modulation of miR-21-5p by mimics and inhibitors also reduced *SMAD7* and *STAT3* expression, however, this did not affect the deleterious effects of GLT challenge on the 1.1B4 beta cells. We suggest that examination of the networks of miRNA pathways regulated by HDL is necessary to understand the epigenetic regulation required to simulate HDL protection of beta cells in developing future therapeutics.

## Figures and Tables

**Figure 1 biology-10-00218-f001:**
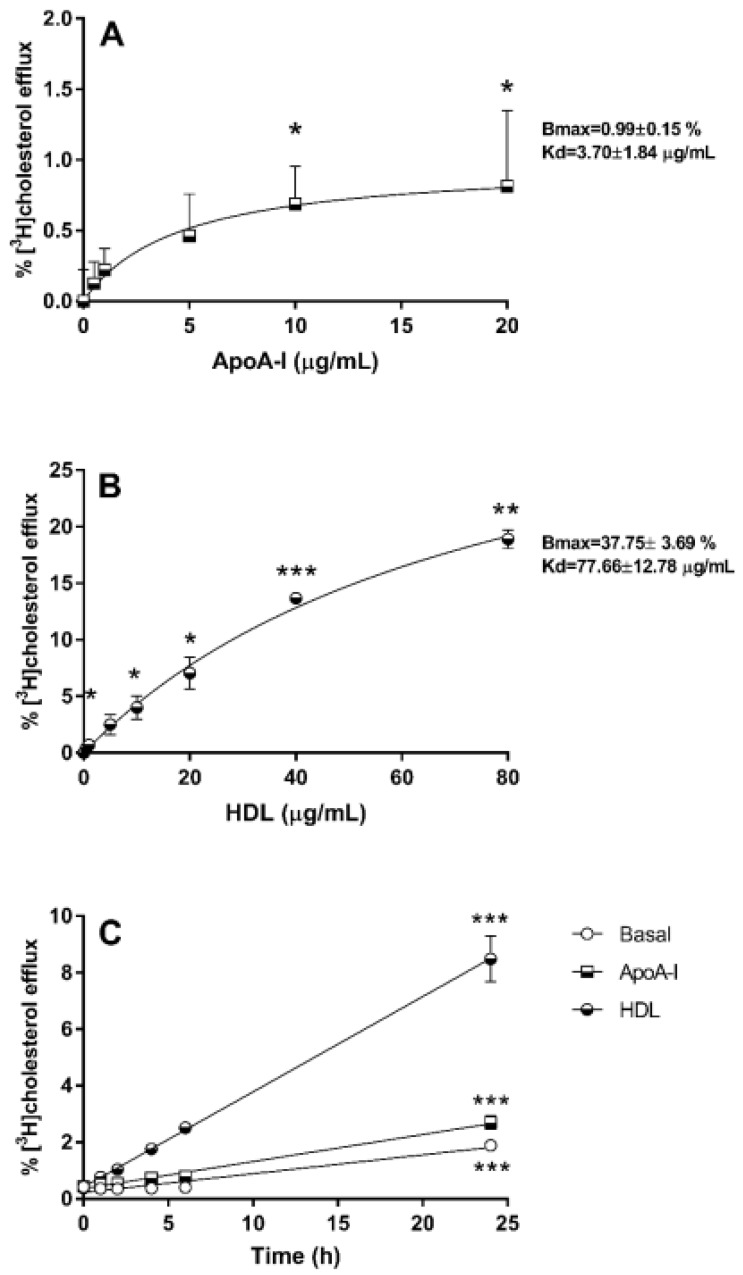
The dose dependencies of [^3^H]cholesterol efflux to apoA-I (24 h) and HDL (24 h) are shown in (**A**,**B**), respectively. The data were analysed using repeated measures ANOVA with post hoc test (Dunnett’s) compared to the 0 μg mL^−1^ measurement; * *p* < 0.05, ** *p* < 0.01 and *** *p* < 0.001. Values are the mean ± SD of three independent experiments (each performed in triplicate). The time dependency of [^3^H]cholesterol efflux under basal conditions, and to apoA-I (20 μg mL^−1^) and HDL (20 μg mL^−1^) is shown in (**C**). The data was analysed using two-way ANOVA with post hoc test (Dunnett’s) compared to the 0h timepoint; * *p* < 0.05, *** *p* < 0.001. Values are the mean ± SD of between three and four independent experiments (each performed in triplicate). Standard deviations, instead of SEM, have been shown, as the SEM for (**B**,**C**) are too small to be viewed.

**Figure 2 biology-10-00218-f002:**
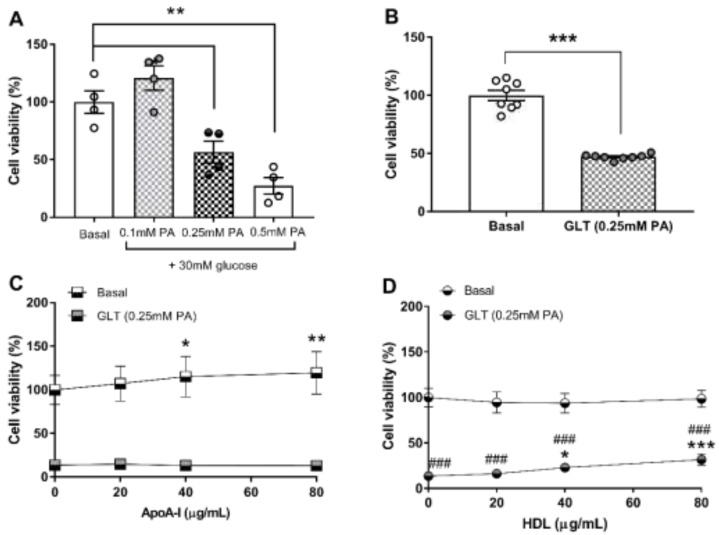
The impact of palmitic acid (PA; 0.1 mM to 0.5 mM), in the presence of 30 mM glucose (GLT challenge), on the viability of 1.1B4 cells, as judged by conversion of MTT to formazan in four independent experiments (mean ± SEM), normalised to the basal condition, is shown in (**A**). Data were analysed by one-way ANOVA with post hoc tests compared to the basal group * *p* < 0.05, ** *p* < 0.01. The effect of 0.25 mM PA, under the same conditions, measured by conversion of resazurin to fluorescent resorufin, is shown in (**B**); values are the mean ± S.D. of eight independent replicates in a single experiment; the data were analysed by Students’ *t*-test; *** *p* < 0.001. The impact of apoA-I (0–80 μg mL^−1^) on the viability of 1.1B4 cells under GLT challenge is shown in (**C**) (mean ± SEM; *n* = 3); the data were analysed by two-way ANOVA with post hoc test (Dunnett’s) compared to the basal group in the absence of apoA-I * *p* < 0.05, *** *p* < 0.001. The effect of HDL (0–80 μg mL^−1^) under the same conditions is shown in (**D**) (mean ± SEM; *n* = 3). The data were analysed by two-way ANOVA with post hoc test (Dunnett’s) compared to the basal group in the absence of HDL. ### *p* < 0.001 compared with the GLT challenge alone, or *** *p* < 0.001 compared with the GLT challenge alone.

**Figure 3 biology-10-00218-f003:**
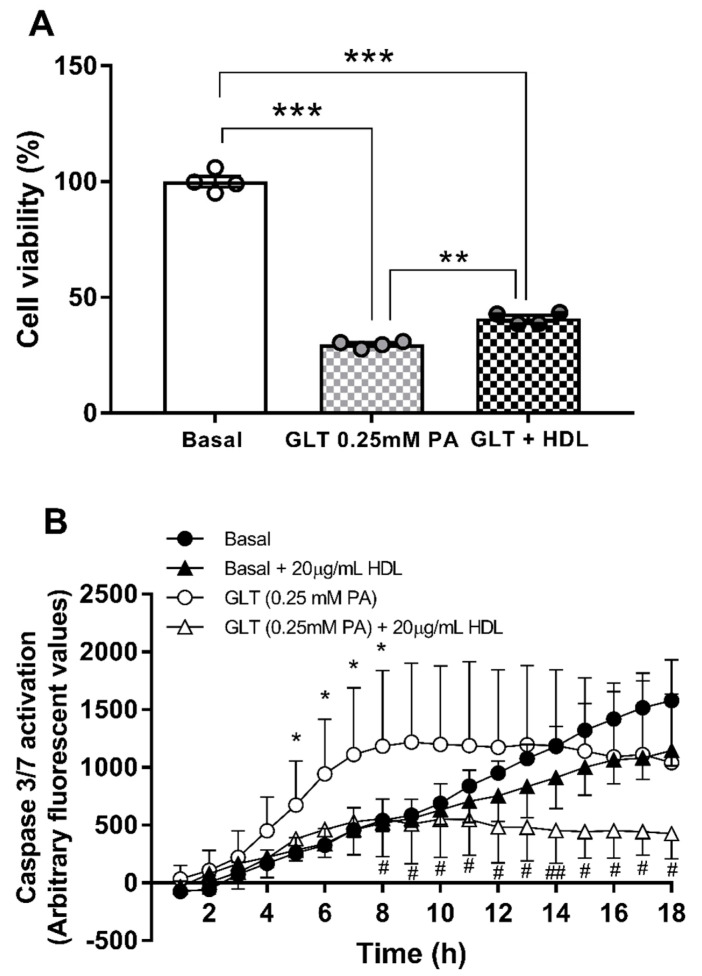
The effect of HDL (20 μg mL^−1^) on cell viability, tested by resorufin conversion, under GLT challenge (0.25 mM PA and 30mM) glucose is shown in (**A**); results are the mean ± SEM of 4 independent experiments, analysed by one-way ANOVA with post hoc (Dunnett’s) test for the comparisons indicated: ** *p* < 0.01 and *** *p* < 0.001. The ability of the same concentration of HDL to protect against GLT-induced apoptosis in 1.1B4 cells, measured by caspase3/7 activation, is shown in (**B**); values are the mean ± SEM of 4 independent replicates in a single experiment. Statistical analysis was performed using repeated measures ANOVA with post hoc (Dunnett’s) test compared to the basal condition; * *p* < 0.05 and compared to GLT challenge alone: # *p* < 0.05 and ## *p* < 0.01 compared at the time points indicated.

**Figure 4 biology-10-00218-f004:**
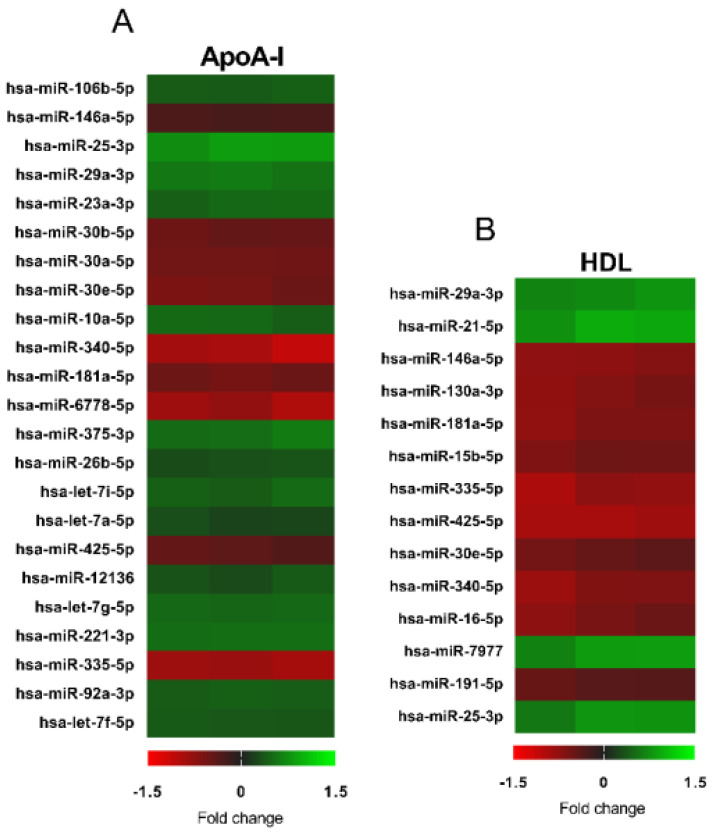
Heatmaps depicting miRNAs that are differentially expressed following a 24 h treatment with 20 µg mL^−1^.ApoA-I (**A**) or 20 µg mL^−1^ HDL (**B**). Data are presented from 1 sample per group each with an independent microchip array containing three technical replicates per sequence on each microchip. All miRNAs presented show differences *p* < 0.05 (Students’ *t*-test) compared to the basal control, and a signal >500. The miRNAs are ranked in order of highly statistically significant miRNAs.

**Figure 5 biology-10-00218-f005:**
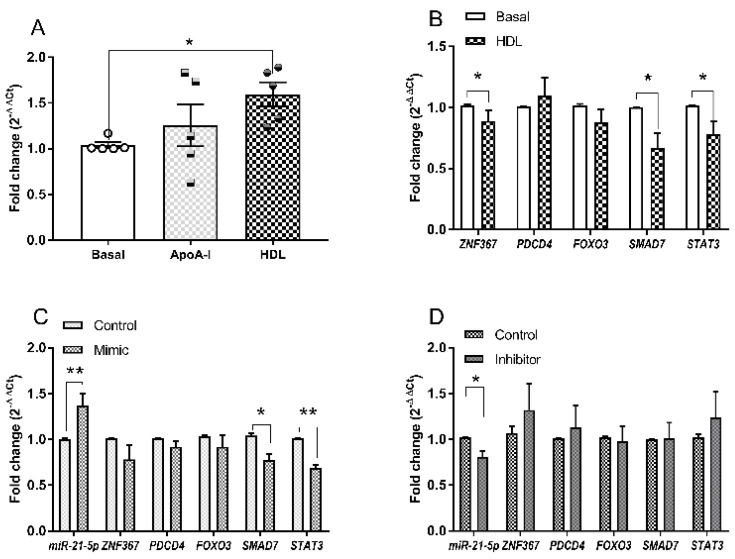
Levels of miR-21-5p relative to RPL13A in 1.1B4 cells, after treatment (24 h) with apoA-I (20 µg mL^−1^) and HDL (20 µg mL^−1^) are shown in (**A**); values are mean ± SEM of 5 independent experiments, analysed by one-way ANOVA with post hoc (Dunnett’s) test; * *p* < 0.05. Gene expressions of the miR-21-5p targets indicated (relative to PPIA), after 24 h treatment with HDL (20 µg mL^−1^) are shown in (**B**); values are mean ± SEM of five to eight independent experiments, analysed by Student’s *t*-test for the comparators indicated; * *p* < 0.05. Expression of hsa-miR-21-5p relative to RPL13A in 1.1B4 cells after transfection with mimic and inhibitor, together with the indicated gene targets, relative to PPIA, are shown in (**C**,**D**), respectively; values are the mean ± SEM of five to seven independent experiments. Statistical analysis was carried out by Students’ *t*-test for the comparators indicated; * *p* < 0.05 and ** *p* < 0.01.

**Figure 6 biology-10-00218-f006:**
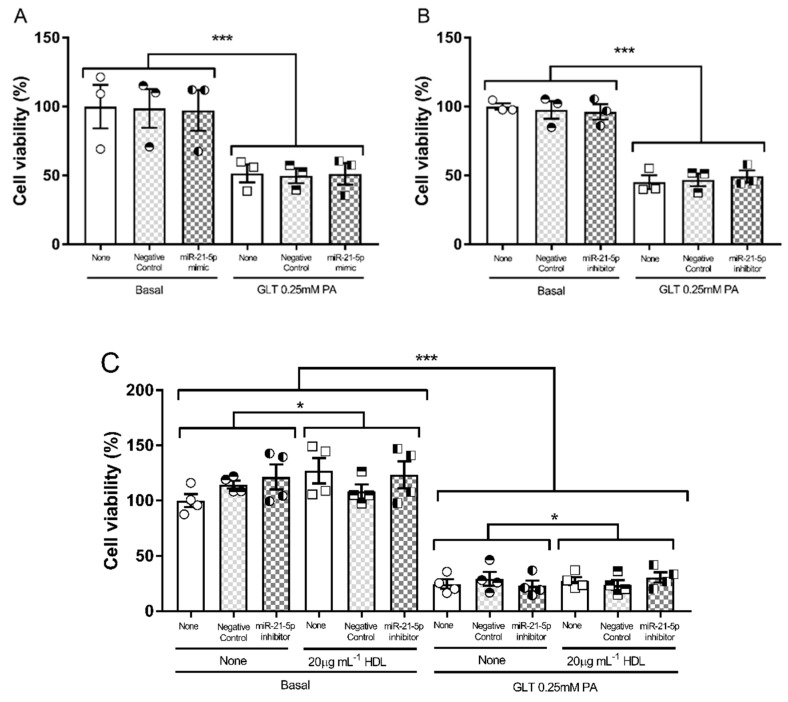
The influence of 5 nM miR-21-5p mimic or 10 nM miR-21-5p inhibitor on the response of 1.1B4 cells to GLT challenge was assessed by conversion of MTT to formazan, normalized to the basal condition, is shown in (**A**,**B**), respectively. This showed that while there was a significant effect of GLT challenge on cell viability in both figures, neither the miR-21-5p mimic or inhibitor modulated the impact of GLT change on the 1.1B4 cells; values are mean ± SEM of three independent experiments, analysed by two-way ANOVA and post hoc (Dunnett’s) test *** *p* < 0.001. The effect of the miR-21-5p inhibitor in the presence of HDL (20 μg mL^−1^) is shown in (**C**). The GLT challenge adversely affected cell viability and there was a significant protective effect of HDL; mean ± SEM of four independent experiments, analysed by three-way ANOVA and post hoc (Dunnett’s) test; * *p* < 0.05 and *** *p* < 0.001.

**Figure 7 biology-10-00218-f007:**
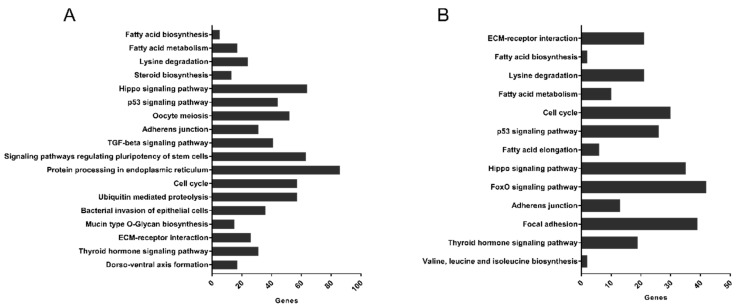
The KEGG pathways strongly associated with the miRNAs differentially expressed following 20 µg mL^−1^ HDL (24 h) treatment were identified through bioinformatics analyses using miRPath v.3.0. Ten downregulated miRNAs and 4 upregulated miRNAs were used for analysis. Pathways associated with downregulated or upregulated miRNAs are presented in (**A**,**B**), respectively. Pathways are ranked, in order of the most closely associated pathways.

**Figure 8 biology-10-00218-f008:**
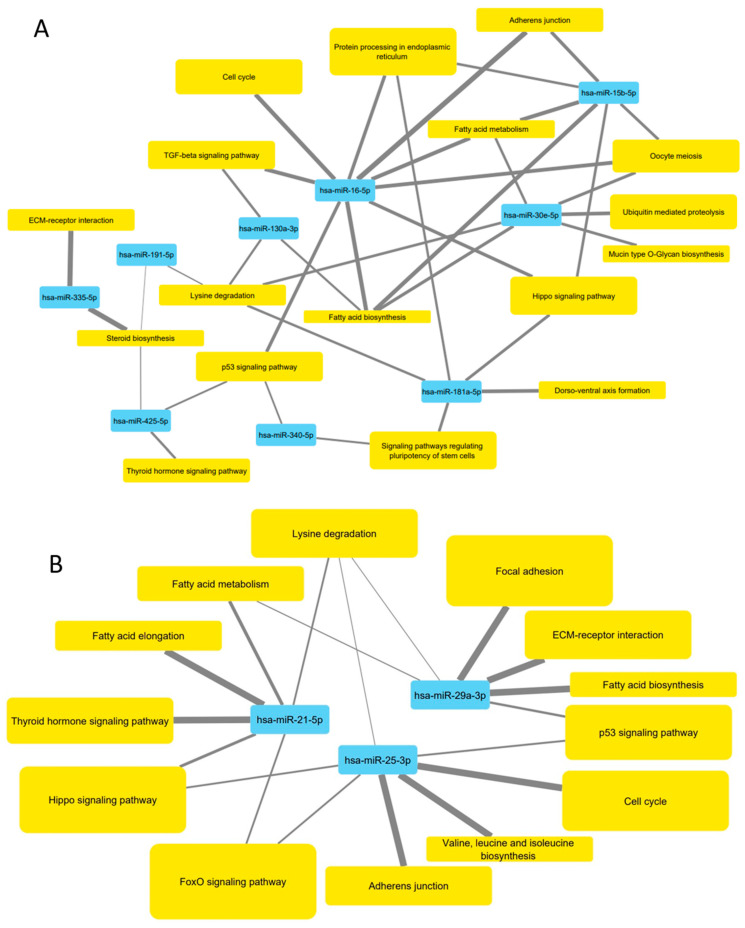
Interaction networks between miRNAs differentially expressed following 20 µg mL^−1^ HDL treatment and their associated pathways were produced in Cytoscape v3.8.0. Ten downregulated miRNAs and 4 upregulated miRNAs were used for analysis. The interaction networks depicting miRNA-pathway connections for downregulated and upregulated miRNAs are presented in (**A**,**B**), respectively. The networks are annotated so that the size of the pathway labels are proportional to the number of genes in the network associated with the relevant miRNAs and the thickness of the edges corresponds to the proportion of the total miRNA:gene interactions in each pathway.

## Data Availability

The data presented in this study are openly available in Mendeley Data at (doi:10.17632/s3bznmgtpg.1).

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
