# Peer review of "Protection against Glucolipotoxicity by High Density Lipoprotein in Human PANC-1 Hybrid 1.1B4 Pancreatic Beta Cells: The Role of microRNA"

_biology, 2021, doi:10.3390/biology10030218_

Round 1
Reviewer 1 Report
It has been revised mainly according to comments. However, the used Human PANC-1.1B4/human beta cell hybrid 1,1B4 cells (1.1B4 cells) were discontinued by ECACC. It means that reproducible experiment is impossible, and this limitation shall be added in addition.
Reviewer 2 Report
Unfortunately, the authors were unable address the main point of criticism of their study satisfactory in the revision of the manuscript. The cardinal mistake that the authors made was the use of the cell line 1.1B4 as a model for a supposedly human pancreatic beta cell line cannot be completely cured. According to information from the ECACC (https://www.phe-culturecollections.org.uk/
products/celllines/discontinued-ecacc-cell-lines/11b4.aspx) the cell line is no longer available because it is contaminated with rat DNA. This means that the cells are only partially of human origin. This also raises doubts as to the extent to which the cell mixture actually exhibits properties of pancreatic beta cells. Thus, the validity of the study, which is already limited, is completely called into question. Unfortunately, the authors failed to carry out genetic analyzes of the cell line themselves in order to be able to estimate the degree of contamination and to at least partially refute this objection. The problem of contamination of the 1.1B4 cells cannot be solved by citing studies that were published by other groups in the years 2019 and 2020. The limited informative value of the study, which mechanistically does not provide any new findings, is also acknowledged by the authors themselves at the end of the discussion section.
This manuscript is a resubmission of an earlier submission. The following is a list of the peer review reports and author responses from that submission.
Round 1
Reviewer 1 Report
High density lipoproteins provide protection against the damaging of glucolipotoxicity in pancreatic beta cells and this submission investigated the epigenetic changes in small non-coding microRNA sequences induced by high density lipoproteins. I like to give the following comments.
- Human PANC-1.1B4 cells used in the present study needs background(s).
- Results expressed as means ± SD or SEM seem unusual. Please show it for one as regular.
- Data in Figure 1A showed a small change in the percentage of cholesterol efflux. Please add the reference(s) to support it.
- What is the change in insulin level by glucolipotoxicity in this cell line?
- The preparation of graphical abstract remained unclear. Please indicate it in detail. Similar to the interaction networks.
- The novel suggestion that networks of regulatory microRNA sequences are more important than single sequences. But, how to do it during application? Please discuss it in clear.
- Modulation of miR-21-5p did not affect the deleterious effects of GLT challenge. Is it same for insulin secretion?
- Limitation(s) of the present study did not discuss. Why?
Reviewer 2 Report
There is some evidence in the literature that HDL can provide protection against glucolipotoxicity in beta cells, which contributes to in type 2 diabetes manifestation. The authors figured out that in the beta cell line 1.1B4 HDL is able to sustain linear cholesterol efflux and can increase concentration-dependent the cell viability under GLT culture conditions. With a resazurin assay, an increase in viability through HDL treatment under GLT conditions from 30 to 40% was demonstrated. A microarray-based analysis of the microRNA expression in the cells after ApoA-I or HDL incubation resulted in identification of sequences common to both treatments. a number of miR sequences appear regulated selectively by HDL treatment, including increased expression of hsa-miR-21-5p and hsa-miR-7977, and loss of expression of hsa-217 miR-15b-5p, hsa-miR-16-5p, hsa-miR-130a-3p and hsa-miR-191-5p. The elevation of hsa-miR-21-5p in 1.1B4 cells detected by microchip analysis was confirmed by Q-PCR. Transfection with the control, mimic or inhibitor of hsa-miR-21-5p did not alter cell viability compared with the wild type cells. Bioinformatic analysis showed that 18 KEGG pathways were associated with miRNAs downregulated HDL incubation and 13 KEGG pathways were associated with upregulated miRNAs. Modulation of miR-21-5p by 372 mimics and inhibitors also reduced SMAD7 and STAT3 expression, however, this did not affect the 373 deleterious effects of GLT challenge on the 1.1B4 beta cells.
I have concerns that the manuscript will be published in its current form. Please address the following point of criticism.
Major points of criticism
According to the website of the Public Health England Culture Collections the human beta cell line 1.1B4 contains rat DNA and is no longer available from ECACC. The International Authentication Cell Line Commitee have been informed.
https://www.phe-culturecollections.org.uk/products/celllines/discontinued-ecacc-cell-lines/11b4.aspx
The use of this contaminated cell line fundamentally calls into question the quality and value of the data collected.
The authors should explain why they chose to perform detailed analysis on has-miR-21-5p and not on other miRs. The importance has-miR-21-5p for GLT is rather minor.
Did the authors check the stability of the reference genes PPIA and RPL13a after treating the cells with HDL? Please add this data in the supplemental. Why used the authors two different reference genes? Please consider the application of the MIQE guidelines for the implementation and publication of qPCR data (The MIQE guidelines: minimum information for publication of quantitative real-time PCR experiments. Clin Chem. 2009 Apr;55(4):611-22.doi: 10.1373/clinchem.2008.112797).
Minor points of criticism
The authors presented in line 196/197 ABCG4 and ABCA1 gene expression data but did not discuss the relevance these values. Was the efficiency of the ABCG4 and ABCA1 qPCR primers determined to make the expression level comparable between the different genes.
The authors should explain why they decided to investigate has-miR-21-5p and not other miR more detail. The importance for GLT is rather minor.
Why did the authors present the formazan formation rate as absolute values? This procedure does not correspond to the procedure described in literature [30] cited by the authors. It is also not clear to me how the authors determine the formation rate. To my opinion this requires the use of a standard. Usually relative values normalized to the control are shown.
